# Impact of pregabalin reclassification as a controlled substance in Egypt on gabapentinoid and opioid utilization: A repeated cross-sectional study

Araniy Santhireswaran[1]*, Simon LaRue[2], Shanzeh Chaudhry[1], Yilei Liu[3], Katie Suda[4], Omneya Mohamed[5], Mohamed Amin[6], Jason Guertin[2,7], Mina Tadrous[1]

**1** Leslie Dan Faculty of Pharmacy, University of Toronto, Toronto, Ontario, Canada, **2** Axe santé des populations et pratiques optimales en santé, Centre de recherche du CHU de Québec-Université Laval, Quebec, Quebec, Canada, **3** Department of Epidemiology, School of Public Health, University of Pittsburgh, Pittsburgh, Pennsylvania, United States of America, **4** VA Pittsburgh Healthcare System and Division of General Internal Medicine, Center for Health Equity Research and Promotion, University of Pittsburgh Schools of Medicine and Pharmacy, Pittsburgh, Pennsylvania, United States of America, **5** Head, Health Economic, Market Access and Reimbursement (HEMAR), Middle East and Africa, Baxter Renal Care, Dubai, United Arab Emirates, **6** Faculty of Pharmacy, Alamein International University, Alamein, Egypt, **7** Département de médecine sociale et préventive, Université Laval, Quebec, Quebec, Canada

* araniy.santhireswaran@mail.utoronto.ca

## Abstract

### Background

Pregabalin is commonly used for treating pain but is also recognized for its misuse potential. In response to rising abuse, Egyptian health authorities reclassified pregabalin as a controlled substance in August 2019, aiming to curb misuse and regulate its distribution. This study evaluated the impact of the 2019 policy on gabapentinoid (pregabalin and gabapentin) and opioid sales in Egypt.

### Methods

An interrupted time-series analysis using Autoregressive Integrated Moving Average (ARIMA) models was conducted on IQVIA MIDAS® quarterly volume sales data obtained under license from IQVIA for the period 2012–2023. Copyright IQVIA. All rights reserved. Drug volume sales were standardized per 1,000 individuals based on population estimates. ARIMA modelling was used to capture immediate and delayed effects of the August 2019 policy change. Percent changes for 3-, 6- and 12-months were also calculated.

### Results

Overall gabapentinoid sales increased steadily until the second quarter of 2019. Following reclassification, a significant decline in total gabapentinoid sales (−67%) was

**Data availability statement:** Data used in the study was third-party global drug purchasing data available from IQVIA. Data cannot be shared publicly because it is non-proprietary and was obtained at a cost from IQVIA. Data can be obtained by contacting IQVIA through their website (https://www.iqvia.com/contact), no special privileges were received in accessing data, and others would able to access data at a cost in the same manner.

**Funding:** The author(s) received no specific funding for this work.

**Competing interests:** The authors have declared that no competing interests exist.

observed, driven by a 99% drop in pregabalin sales, while gabapentin sales surged by 198%. ARIMA analyses of gabapentinoid sales showed a significant short-term effect (pulse: p < 0.001) and a notable gradual long-term change (ramp: p = 0.008). In contrast, opioids exhibited a significant short-term sustained increase (step: p = 0.010) but a non-significant gradual long-term change (ramp: p = 0.256), with sales rising by up to 49.5% at one year post-policy.

## Conclusions

Reclassifying pregabalin effectively reduced its utilization but prompted a shift to gabapentin use. Our findings highlight the complexity of drug policy interventions, underscoring the need for continuous monitoring to mitigate unintended substitution effects and better understand policy impacts of the treatment of pain.

## Introduction

Pregabalin is a gamma-aminobutyric acid analogue and prodrug of gabapentin, commonly used to treat neuropathic pain [1, 2]. In recent years, pregabalin has been associated with a high potential for abuse and has emerged as an illicit drug [3]. Literature reviews have highlighted its euphoria-inducing and dissociative effects in users, underscoring its potential for abuse [4, 5]. In November 2018, pregabalin was classified as a psychoactive substance by the WHO for having considerable abuse liability [6]. Moreover, pregabalin is often taken simultaneously with other opioids which has been associated with a significant increase in mortality [7].

The misuse of pregabalin is becoming increasingly prevalent worldwide with many cases being reported by Scandinavian, British, French, and German pharmacovigilance systems since 2008 [7]. This led to many policies enacted by authorities worldwide to minimize misuse including the United States, Saudi Arabia, France, United Kingdom and New Zealand [8–11]. A notable rise in abuse has been reported in the Middle East, especially in Egypt, where pregabalin misuse ranks fourth among all substances misused, accounting for 30%.[1] In response to this growing issue, Egyptian health authorities added pregabalin to the controlled drug list in August 2019 [12,13]. This required an authorized physician prescription to be provided by the patient or caregiver for the drug to be dispensed at a community pharmacy. The community pharmacy is to keep the prescription so it can be examined by a Ministry of Health if needed. This policy change aimed to restrict its accessibility and regulate its distribution through pharmacies across the country [14]. This study aimed to assess the impact of the August 2019 policy change in Egypt on drug use patterns of gabapentinoids and opioids.

## Methods

### Study design & data source

We used IQVIA MIDAS® quarterly volume sales data for the period Q1 2012 to Q3 2023 to analyse sales of oral and parenteral formulations of pregabalin, gabapentin

and opioids (including codeine, fentanyl, hydromorphone, morphine, oxycodone, tramadol, nalbuphine and pethidine) in Egypt. IQVIA MIDAS reports monthly sales volume data from health systems, retail pharmacies, and major retail outlets (mass merchandisers, grocery stores, and convenience stores without pharmacies) for prescription and over-the-counter (OTC) drugs by country. Sales volume is reported in standardized units, representing the smallest unit of consumption (e.g., one tablet, one vial, or 5 mL of liquid). IQVIA MIDAS integrates IQVIA's national audits to provide a global view of the pharmaceutical market, displaying estimated product volumes of medications through retail and hospital channels. It is widely used in pharmacoepidemiology research and is known for its high accuracy. An internal validation study by IQVIA reports a precision rate of 92.6% for Egypt of wholesaler, distributer and pharmacy audits. This indicates that it captures 92.6% of purchases made by drugstores wholesalers and manufacturers, suggesting sales volume is a good proxy for studying drug utilization [15]. Drug sales were standardized per 1,000 individuals, based on Egypt's population estimates from the 2024 World Population Prospects, calculated as of July 1 each year [16]. The analysis focused on retail sales, as opioid use patterns differ between hospital and community settings, allowing for a more uniform comparison. Because the analysis was restricted to a single jurisdiction and comparisons were made within drug classes, standardized units provide a reliable measure of drug use. Standardized dose metrics such as defined daily doses were not used, as the consistency in formulation strengths and dosing regimens minimizes the potential for inconsistencies in measurements.

## Statistical analysis

To quantify changes in sales following the policy change, 3-, 6- and 12-month percent changes were calculated. See S1 Fig for example calculation. An interrupted time series analysis was performed using Autoregressive Integrated Moving Average (ARIMA) modeling for each drug class. ARIMA modeling is a method used to assess the impact of an intervention on longitudinal data. Compared with other interrupted time-series methods, the ARIMA framework accounts for autocorrelation and seasonality in the outcome time series, providing more reliable estimates when there is correlation between time points. Given the seasonality and autocorrelation seen in drug sales data the ARIMA framework is better suited for this analysis [17]. To evaluate the delayed effects of the August 2019 policy change, a ramp function was incorporated into the ARIMA models, starting in the third quarter of 2019. Step function was used for opioids to account for immediate sustained changes, and a pulse function was used for gabapentinoids to account for immediate temporary changes, reflecting whether these changes persisted or returned to pre-intervention levels. ARIMA model intervention coding is specified in the supplementary material (S1 Table). The data for the overall gabapentinoid ARIMA model was transformed to the log scale in order to lessen heteroscedasticity in the time series. Seasonality and non-stationarity were addressed through differencing, as recommended by the Kwiatkowski unit root test (S2 Table)[18]. Autoregressive and moving average terms were selected based on autocorrelation function and partial autocorrelation function plots (S2 Fig). The optimal model fit was determined using the Ljung-Box test for white noise and fit statistics, including the Akaike Information Criterion corrected (AICc) and Bayesian Information Criterion (BIC). Key statistics for model fit are provided in the supplementary material (S3 Table). All analyses were conducted using R version 4.4.1 [19].

## Results

Overall wholesale sales of gabapentinoid per 1,000 individuals gradually increased between 2012 and 2019, peaking in quarter 2 of 2019 before the intervention (Fig 1). Before the policy change, overall sales of gabapentinoid gradually increased, ranging from 163 to 1910 units per 1000 individuals, averaging 555 units per 1000. This rise was driven by increased pregabalin sales ranging from 94 to 1769 units, while gabapentin sales remained stable at 65–141 units. In the quarter following the policy change, a sudden decline of 67.0% in overall gabapentinoid volume sales was observed, which shifted to a 31.7% decrease at six months and a 24.4% decrease at one year (Table 1). This pattern was primarily driven by the sharp drop in pregabalin volume sales of −99.0%, −97.8%, and −97.6% at three, six, and twelve months, respectively, and a concurrent surge in gabapentin sales rising by 198.5%, 516.6%, and 583.0% over the same intervals

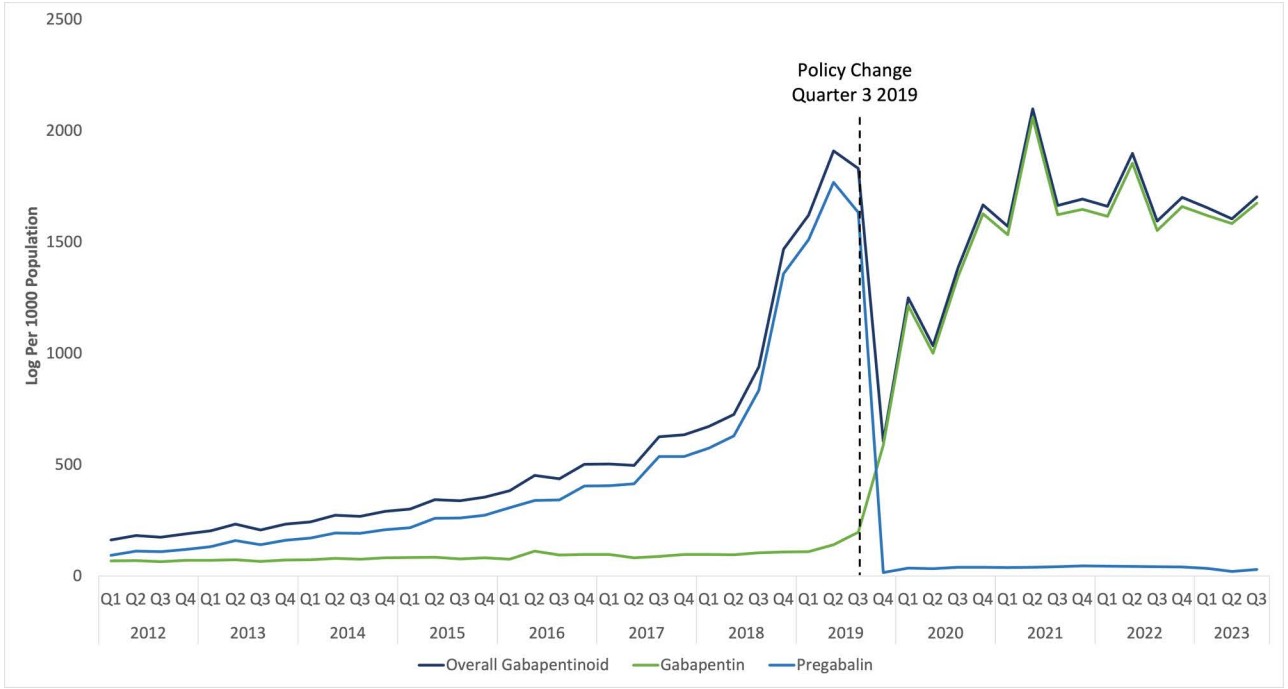

**Fig 1. Gabapentinoid sales per 1000 population from 2012–2023.** Author analysis based on IQVIA MIDAS® quarterly volume sales data for Egypt for the period Q1 2012 to Q3 2023, reflecting estimates of real-world activity. Copyright IQVIA. All rights reserved.

**Table 1. Percent changes and population adjusted sales volume per 1000 of gabapentinoid and opioid sales at 3-months, 6-months and 12-months following policy change.**

| Drug Class | Pre-Intervention Population Adjusted Sales | Post-Intervention Percent Changes (Population Adjusted Sales) | | |
|---|---|---|---|---|
| | | 3-months | 6-months | 12-months |
| **Overall Gabapentinoid** | 1831 | −67.00% (604) | −31.71% (1250) | −24.35% (1385) |
| **Gabapentin** | 197 | +198.49% (588) | +516.57% (1214) | +583.00% (1345) |
| **Pregabalin** | 1634 | −99.00% (16) | −97.81% (36) | −97.57% (40) |
| **Overall Opioid** | 6.12 | +8.10% (6.61) | +37.23% (8.40) | +49.47% (9.15) |

Author analysis based on IQVIA MIDAS® quarterly volume sales data for Egypt for the period Q1 2012 to Q3 2023, reflecting estimates of real-world activity. Copyright IQVIA. All rights reserved.

(Table 1). After the policy change, overall sales of gabapentinoid increased ranging from 604 to 2098 per 1000, averaging 1549 per 1000. A significant pulse trend was observed following the intervention (p < 0.001), and the change in the ramp function after the third quarter of 2019 was also significant (p = 0.008) (Table 2). Counterfactual scenarios for gabapentinoid analysis are shown in Fig 2.

For opioids, data from the first quarter of 2012 was excluded due to unusually high tramadol sales. This may have attributed to tramadol being placed under national control in 2012 [20,21]. Overall, opioid sales per 1,000 population remained stable between 2012 and 2019 (Fig 3). After the intervention period, consumption increased and persisted until quarter 3 of 2020. Overall opioid sales increased by 8.1%, 37.2% and 49.5% at three, six, and twelve months, respectively (Table 1). Opioid volume sales ranged between 4.80 and 6.88 per 1000 individuals, with an average of 6.03 per

**Table 2. ARIMA model coefficients, p-values and Ljung-Box test results for overall gabapentinoids.**

| Model: Intercept+Ramp+Pulse+ARIMA (0, 1, 0)(0, 0, 2) | | | | |
|---|---|---|---|---|
| | Coefficient | Standard Error | 95% Confidence Interval | p-value |
| **Intercept** | 0.095 | 0.029 | 0.039, 0.151 | < 0.001 |
| **Ramp** | −0.115 | 0.043 | −0.199, −0.030 | 0.008 |
| **Pulse** | −0.894 | 0.065 | −1.022, −0.766 | < 0.001 |
| **Seasonal MA1** | −0.310 | 0.130 | −0.565, −0.054 | 0.017 |
| **Seasonal MA2** | 0.739 | 0.355 | 0.043, 1.435 | 0.038 |

**sigma²**: 0.014.

**Ljung-Box Test**: Q*= 2.991, degrees of freedom = 6, p-value = 0.810.

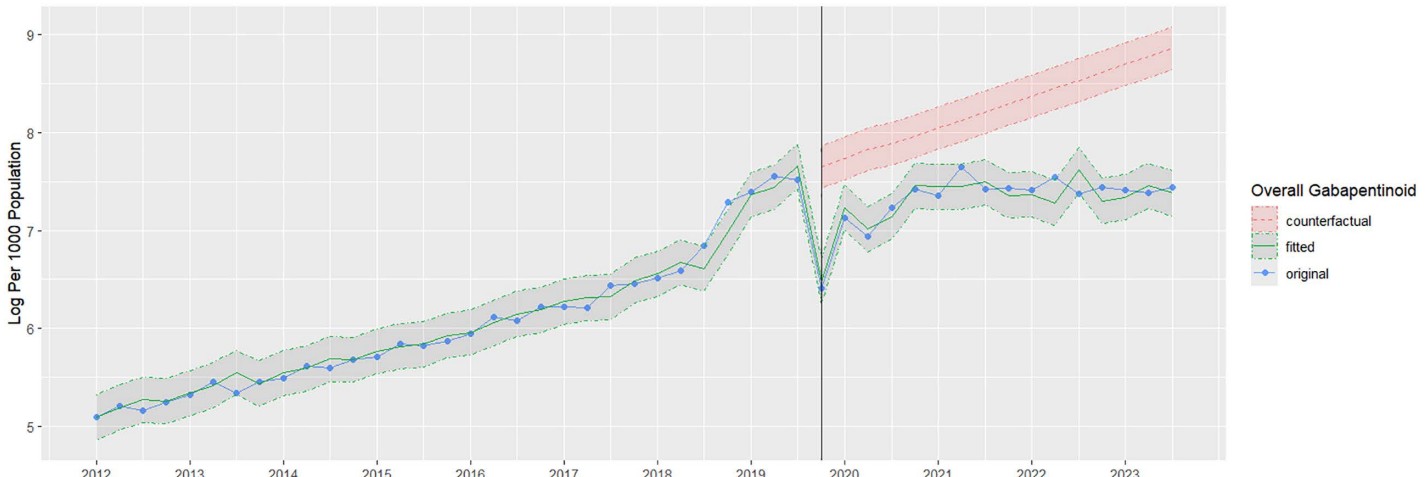

**Fig 2. Overall gabapentinoid sales log per 1000 population with counterfactual scenarios by dashed lines and 95% CI for the post-intervention.** Author analysis based on IQVIA MIDAS® quarterly volume sales data for Egypt for the period Q1 2012 to Q3 2023, reflecting estimates of real-world activity. Copyright IQVIA. All rights reserved.

1000 before the intervention, whereas post-intervention, it ranged from 4.11 to 9.15 units per 1000, averaging 7.38 units per 1000. The step function in opioid volume sales following the new policy was significant (p=0.010). However, a gradual shift in a ramp function did not achieve significance (p=0.256) (Table 3). Counterfactual scenarios for opioid analysis are shown in Fig 4.

## Discussion

Our results indicated a sharp decline in pregabalin sales post-policy accompanied by a notable increase in gabapentin utilization, suggesting a substitution effect. For opioids, there was an increase in volume sales immediately post-policy. These observations show both similarities and differences compared with evidence from other countries, and comparative analyses across diverse health systems may provide additional insights into the impact of such policies. Furthermore, the changes in opioid and gabapentin sales raise concerns regarding potential concomitant usage, underscoring the need for ongoing monitoring to address associated risks.

Comparing our findings with those from other countries reveals both similarities and variations in the outcomes of pregabalin policy interventions. For instance, a study in Saudi Arabia evaluating the 2018 classification of pregabalin as a controlled

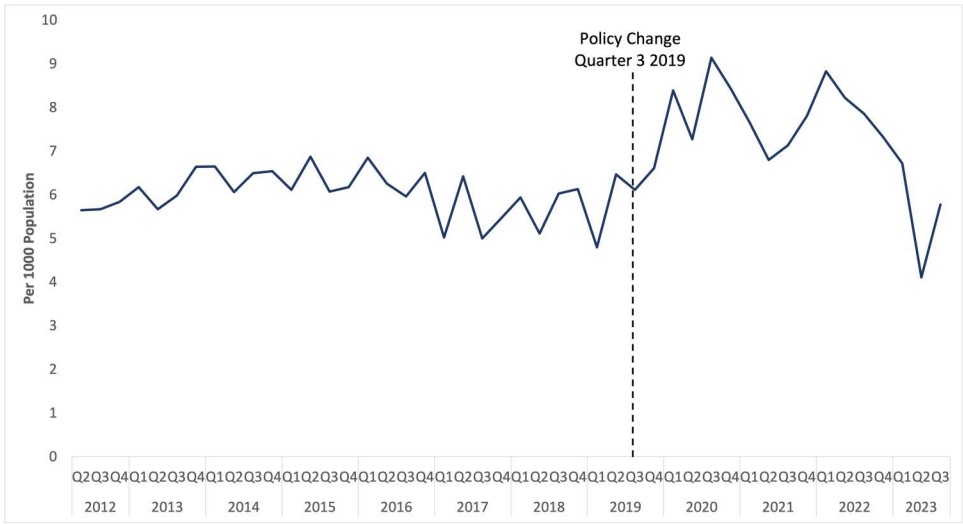

**Fig 3. Overall opioid sales per 1000 population for 2012 to 2023.** Author analysis based on IQVIA MIDAS® quarterly volume sales data for Egypt for the period Q1 2012 to Q3 2023, reflecting estimates of real-world activity. Copyright IQVIA. All rights reserved.

**Table 3. ARIMA model coefficients, p-values and Ljung-Box test results for overall opioids.**

| Model: Intercept + Ramp + Pulse + ARIMA (1, 0, 0)(1, 0, 2) | | | | |
|---|---|---|---|---|
| | Coefficient | Standard Error | 95% Confidence Interval | p-value |
| **Intercept** | 6.110 | 0.221 | 5.678, 6.542 | < 0.001 |
| **Ramp** | −0.055 | 0.049 | −0.151, 0.040 | 0.256 |
| **Step** | 1.569 | 0.607 | 0.379, 2.759 | 0.010 |
| **Non-seasonal AR1** | 0.346 | 0.182 | −0.010, 0.702 | 0.057 |
| **Seasonal AR1** | −0.582 | 0.223 | −1.019, −0.145 | 0.009 |
| **Seasonal MA1** | 0.463 | 0.229 | 0.015, 0.911 | 0.043 |
| **Seasonal MA2** | 0.643 | 0.258 | 0.138, 1.148 | 0.013 |

**sigma²**: 0.449.

**Ljung-Box Test**: Q*= 1.495, degrees of freedom = 4, p-value = 0.828.

substance reported a 21.5% reduction in the total number of pregabalin users but a 15.5% increase in the total number of pregabalin prescriptions. Additionally, prescriptions for tramadol and acetaminophen/codeine rose by 21% and 16.1%, respectively, among these pregabalin users, suggesting a potential shift toward alternative analgesics or co-analgesics [8]. Conversely, in France, where pregabalin was reclassified as a controlled substance in 2021, there was an immediate significant drop of 38,475,375 mg in total pregabalin dispensed, accompanied by a concurrent decrease in co-dispensed opioids [9]. These cross-national differences may reflect variations in regulatory enforcement, prescribing habits, and the availability of alternative pain management options, underscoring the importance of country-specific approaches when implementing drug control policies.

The findings suggest that Egypt's 2019 policy change effectively reduced pregabalin utilization, but the concurrent increase in gabapentin use indicates a substitution effect, where individuals may have switched to an alternative gabapentinoid. The increase in opioid use post-policy suggests potential unintended consequences, possibly due to prescribers or individuals seeking alternative pain management options. However, the lack of a significant change in long-term trends in opioid consumption indicates that the policy did not lead to sustained increases. This may be attributed to existing

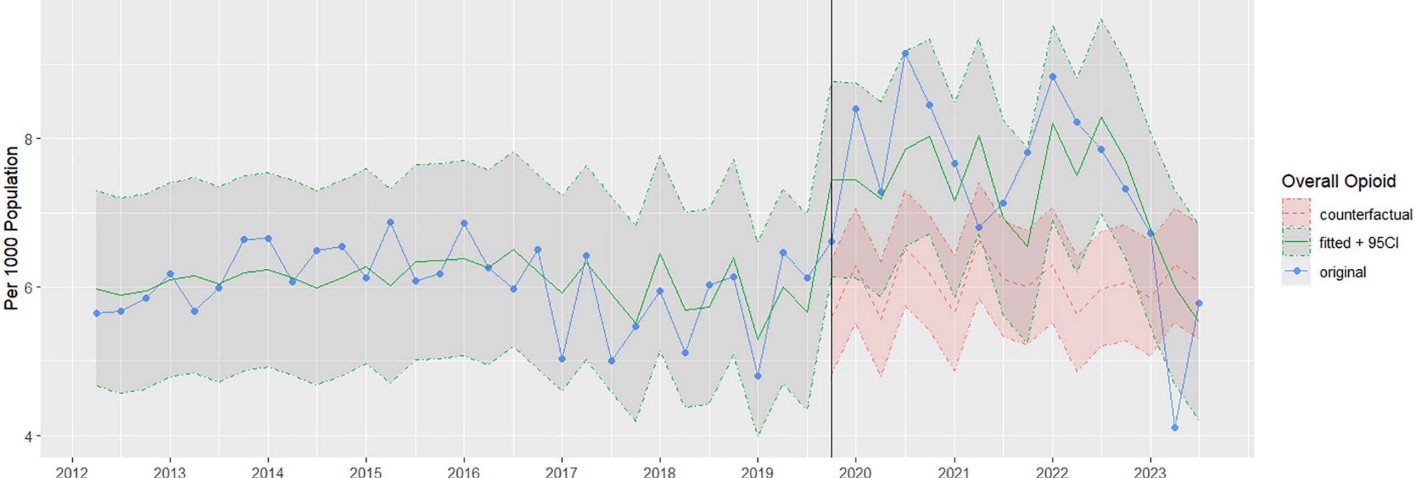

**Fig 4. Overall opioid sales per 1000 population with counterfactual scenarios by dashed lines and 95% CI for the post-intervention.** Author analysis based on IQVIA MIDAS® quarterly volume sales data for Egypt for the period Q1 2012 to Q3 2023, reflecting estimates of real-world activity. Copyright IQVIA. All rights reserved.

opioid regulations or differences in prescribing practices, such as The Egyptian Narcotics Control Law that limits morphine prescribing [22,23]. These findings highlight the complex nature of drug policy interventions, emphasizing the need for continuous monitoring to mitigate substitution effects and prevent unintended shifts in drug use behavior.

This study has several limitations that should be considered. The use of IQVIA MIDAS volume sales data may not fully capture individual-level use behaviors, illicit drug use, or self-medication trends outside formal healthcare settings, restricting the effects we were able to capture. Wholesale-level data reflects the flow of medicines from distributor to pharmacies instead of pharmacies to patients, which can reflect a delayed impact of policy changes and an underestimated decline in drug use. Moreover, because we only licensed value and volume sales data and not patient-level data, we were only able to capture population-level trends, making it difficult to determine whether individuals who reduced pregabalin use switched to gabapentin or opioids. Additionally, this study used an ecological interrupted time series design, which evaluates aggregate-level trends in drug utilization over time rather than individual-level behavior. While this approach is robust for assessing policy impacts, it relies on several key assumptions, including a stable underlying population, no concurrent external interventions, and no time-varying confounders during the study period. Although no other major regulatory or clinical policy changes occurred at the time of the pregabalin policy implementation, it is important to acknowledge these underlying assumptions and the inherent limitations of this design. Lastly, this study focuses on drug volume sales trends rather than clinical outcomes, preventing direct assessment of changes in overdose rates, adverse events, or treatment efficacy. Future research should explore these outcomes to provide a more comprehensive understanding of the policy's long-term effects.

Our findings suggest that policy changes have the potential to be effective when implemented thoughtfully, as demonstrated by the significant decline in pregabalin use following its reclassification. However, our study also suggests that regulating a single drug susceptible to abuse, rather than the entire class, is insufficient to address the problem. To effectively address the substitution effect—where individuals switch from one controlled medication to another within the same class when restrictions are imposed—policy changes can focus on regulating entire drug classes with abuse potential, rather than individual drugs. A gradual implementation of restrictions on a drug class may be a valuable strategy for minimizing disruption allowing patients to work with their prescribers on a gradual transition to treatment alternatives preventing access disruption.

Drug use is influenced by a multifaceted system, and regulatory interventions require continued monitoring to assess ripple effects of policy changes and minimize unintended harm and misuse. Changes in drug policy can lead to

unexpected shifts in prescribing patterns or patient behavior, emphasizing the need for ongoing surveillance to identify and mitigate unintended substitution effects. To ensure that future policies achieve their intended goals without exacerbating other public health risks, regulatory actions should be accompanied by longitudinal assessments to evaluate their broader impact on drug misuse, healthcare utilization, and patient outcomes.

## Supporting information

**S1 Fig. Percent change calculation and example calculation of overall gabapentinoid 3-month post-policy change.**
(DOCX)

**S2 Fig. ACF and PACF plot for overall Gabapentinoid.**
(DOCX)

**S1 Table. ARIMA model intervention coding.**
(DOCX)

**S2 Table. Kwiatkowski unit root test results.**
(DOCX)

**S3 Table. AICc and BIC fit statistics for some models considered.**
(DOCX)

## Acknowledgments

These statements, findings, conclusions, views and opinions contained and expressed herein are not necessarily those of IQVIA.

## Author contributions

**Conceptualization:** Araniy Santhireswaran, Simon LaRue, Yilei Liu, Katie Suda, Omneya Mohamed, Mohamed Amin, Jason Guertin, Mina Tadrous.

**Data curation:** Araniy Santhireswaran, Simon LaRue, Shanzeh Chaudhry.

**Formal analysis:** Araniy Santhireswaran, Yilei Liu.

**Investigation:** Omneya Mohamed, Mohamed Amin, Jason Guertin, Mina Tadrous.

**Methodology:** Araniy Santhireswaran, Simon LaRue, Yilei Liu.

**Project administration:** Araniy Santhireswaran, Simon LaRue, Yilei Liu, Katie Suda, Jason Guertin, Mina Tadrous.

**Supervision:** Katie Suda, Jason Guertin, Mina Tadrous.

**Validation:** Araniy Santhireswaran, Shanzeh Chaudhry, Mina Tadrous.

**Writing – original draft:** Araniy Santhireswaran, Simon LaRue.

**Writing – review & editing:** Simon LaRue, Shanzeh Chaudhry, Yilei Liu, Katie Suda, Omneya Mohamed, Mohamed Amin, Jason Guertin, Mina Tadrous.

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
