## [Decision Letter · Decision Letter 0]

7 Oct 2025

Dear Dr. Santhireswaran,

We look forward to receiving your revised manuscript.

Kind regards,

Claudia Garcia Serpa Osorio-de-Castro, Ph.D

Academic Editor

PLOS ONE

2. For studies involving third-party data, we encourage authors to share any data specific to their analyses that they can legally distribute. PLOS recognizes, however, that authors may be using third-party data they do not have the rights to share. When third-party data cannot be publicly shared, authors must provide all information necessary for interested researchers to apply to gain access to the data. (https://journals.plos.org/plosone/s/data-availability#loc-acceptable-data-access-restrictions)

Additional Editor Comments (if provided):

Reviewers' comments:

Reviewer's Responses to Questions

**Comments to the Author**

1. Is the manuscript technically sound, and do the data support the conclusions?

Reviewer #1: Partly

Reviewer #2: Partly

2. Has the statistical analysis been performed appropriately and rigorously?

Reviewer #1: No

Reviewer #2: No

3. Have the authors made all data underlying the findings in their manuscript fully available?

Reviewer #1: No

Reviewer #2: Yes

4. Is the manuscript presented in an intelligible fashion and written in standard English?

Reviewer #1: Yes

Reviewer #2: Yes

Reviewer #1: The manuscript "Impact of reclassification of pregabalin as a controlled substance in Egypt: a repeated

cross-sectional analysis" addresses and evaluates the impact of an important pharmaceutical policy intervention in Egypt using a quasi-experimental research design.

The authors have used proprietary IQVIA data, which is sales data, to evaluate the policy's impact. Uncontrolled ITS can be considered a reasonable research design to evaluate such policy intervention, given that the data has a sufficient number of pre- and post-intervention data points, which the study had. However, it is important to note that this approach assumes that there are no time-varying confounders, no concomitant policy changes affecting the outcome in question and a stable population. These factors should be adequately reported and discussed in the discussion section. The authors reported that they have taken care of seasonality and non-stationarity by using differencing and selected AR and MA terms using AC and PAC plots, which should be submitted in the supplementary file. Furthermore, the presentation of the results, especially about pre- and post-intervention magnitude changes (descriptive stats) in the table, can be improved significantly by having explicit column headings (pre- and post-intervention) to report the absolute values and % changes. Please revise the

Similarly, a better way to report the ITS (segmented regression) is to fully report the results in a tabular format where the reader can appreciate the baseline (pre-intervention and post-intervention level and trend changes) by looking at coefficients and associated 95% CIs.

Finally, the graph representing the trend changes should also report counterfactual scenarios by dashed lines and 95% CI for the post-intervention trend line.

The authors have succinctly captured the limitations of the study, which is commendable.

Reviewer #2: 1. The term “reclassification” in the title is ambiguous. Since pregabalin was designated as a controlled substance, it would be clearer if the title explicitly reflected this.

2. The title should clearly specify what the “impact” refers to. Please revise the title to indicate that the study evaluates the impact on gabapentinoid and opioid utilization in Egypt.

3. The title specifies “repeated cross-sectional analysis,” but the design is more accurately described as an interrupted time series analysis using ARIMA models.

4. In the abstract, the terms “pulse,” “ramp,” and “step” are presented from a modeling perspective. It would be clearer to replace these with interpretation-oriented terms, such as immediate short-term effect, sustained level change, or gradual long-term trend change.

5. The description of IQVIA’s MIDAS database is insufficient for international readers who may not be familiar with it. More detail is needed on what the database is, how it is constructed, and what types of information it provides. In particular, please explain how drug information is captured and reported in MIDAS, and clarify whether this database can be considered representative for assessing medication use in Egypt. This additional context would help readers evaluate the validity of using MIDAS data for the study objectives.

6. The criteria by which the study data were selected from the MIDAS database should be described in the Methods section. Additionally, the results of this selection should be illustrated in a flow diagram and included in the Results section.

7. In the Methods section, the opioids are described as “including codeine, fentanyl, hydromorphone, morphine, oxycodone, tramadol, and others.” The use of “including” and “others” makes the definition unclear. A complete and explicit list of all opioid medications analyzed should be provided to ensure clarity and reproducibility.

8. The description of the study model in the Methods section is not sufficiently detailed for reproducibility. More specific information on the intervention coding (step, pulse, ramp) and the final ARIMA specifications, including seasonal terms, should be provided. The ITS model should also be described in equation form, if available.

9. The Methods section should explain why ARIMA modeling was chosen over other common ITS approaches, such as segmented regression. Providing the rationale for this methodological choice would help readers better understand the strengths and limitations of the analysis.

10. The exact calculation method for the 3-, 6-, and 12-month percent changes should be provided.

11. It appears that drug utilization was estimated and modeled based on the smallest unit of consumption (e.g., one tablet, one vial, or 5 mL of liquid). It is unclear whether this approach was due to limitations of the MIDAS data or for another specific reason. A clear explanation is needed as to why analyses were not conducted using standardized dose measures (e.g., defined daily dose), and this limitation should be explicitly acknowledged, as it represents a significant constraint of the study.

12. The final selected ARIMA model specifications and results should be explicitly presented in the Results section. Reporting the model orders, key coefficients, and fit statistics (e.g., AICc, BIC, Ljung–Box test) would allow readers to clearly understand the modeling choices and assess the robustness of the findings.

13. In the Results section, please also report the findings for all intervention functions consistently. For gabapentinoids, the step effect is not mentioned, and for opioids, the pulse effect is not described.

14. The manuscript acknowledges that external factors such as concurrent regulatory changes, clinical guidelines, or shifts in prescribing practices could have influenced drug utilization trends. However, it does not explain why these factors were not incorporated into the ARIMA models. For example, the COVID-19 pandemic and other healthcare policies in Egypt may have significantly affected drug use. Without a clear rationale for excluding these variables, it is challenging to attribute the observed changes solely to the pregabalin policy.

15. In the Discussion, the authors state that the observations align with evidence from other countries. However, as noted, the results in France differed from those in Egypt. A more comprehensive discussion is needed to explore the reasons behind these cross-country differences.

16. The statement that the lack of a significant long-term trend in opioid consumption may be attributed to existing opioid regulations or differences in prescribing practices requires a more detailed interpretation supported by references.

**Do you want your identity to be public for this peer review?** For information about this choice, including consent withdrawal, please see our Privacy Policy

Reviewer #1: **Yes: ** Habib Hasan Farooqui

Reviewer #2: No

---

## [Author Response · Author response to Decision Letter 1]

7 Nov 2025

November, 2025

Dr. Claudia Garcia Serpa Osorio-de-Castro, Ph.D

Academic Editor

PLOS ONE

Re: PLOS ONE Decision: Revision required - PONE-D-25-13595

Dear Dr. Osorio-de-Castro,

We are grateful for the suggestions provided by the reviewers and believe these comments have helped strengthen this paper. We have made the suggested changes to the manuscript based on the suggestions from the reviewer’s comments. A point-by-point response to the comments is provided below. In the manuscript with tracked changes we have highlighted all changes made since the original submission.

Sincerely,

Araniy Santhireswaran

Reviewer 1 Comments

The manuscript "Impact of reclassification of pregabalin as a controlled substance in Egypt: a repeated cross-sectional analysis" addresses and evaluates the impact of an important pharmaceutical policy intervention in Egypt using a quasi-experimental research design.

The authors have used proprietary IQVIA data, which is sales data, to evaluate the policy's impact. Uncontrolled ITS can be considered a reasonable research design to evaluate such policy intervention, given that the data has a sufficient number of pre- and post-intervention data points, which the study had. However, it is important to note that this approach assumes that there are no time-varying confounders, no concomitant policy changes affecting the outcome in question and a stable population. These factors should be adequately reported and discussed in the discussion section.

Response: Thank you for this comment. We agree that ITS analysis assumes the absence of time-varying confounders, no concurrent policy changes, and a stable population. These assumptions are inherent to the ecological nature of the study design. While no other major regulatory or policy changes occurred during the intervention period (Q3 2019), we recognize that ecological ITS designs cannot fully rule out the influence of unmeasured external factors. We have revised the discussion section to clarify the assumptions underlying our analytic approach.

Change: Additionally, this study used an ecological interrupted time series design, which evaluates aggregate-level trends in drug utilization over time rather than individual-level behavior. While this approach is robust for assessing policy impacts, it relies on several key assumptions, including a stable underlying population, no concurrent external interventions, and no time-varying confounders during the study period. Although no other major regulatory or clinical policy changes occurred at the time of the pregabalin policy implementation, it is important to acknowledge these underlying assumptions and the inherent limitations of this design.

The authors reported that they have taken care of seasonality and non-stationarity by using differencing and selected AR and MA terms using AC and PAC plots, which should be submitted in the supplementary file.

Response: Thank you for pointing this out. We have included the AC and PAC plots in the supplement now as Figure S2, and the seasonal and non-seasonal AR and MA terms in tables 2 and 3 of the results.

Change:

Table 2: ARIMA model coefficients, p-values and Ljung-Box test results for overall gabapentinoids.

Model: Intercept + Ramp + Pulse + ARIMA (0, 1, 0)(0, 0, 2)

Coefficient Standard Error 95% Confidence Interval p-value

Intercept 0.095 0.029 0.039, 0.151 < 0.001

Ramp -0.115 0.043 -0.199, -0.030 0.008

Pulse -0.894 0.065 -1.022, -0.766 < 0.001

Seasonal MA1 -0.310 0.130 -0.565, -0.054 0.017

Seasonal MA2 0.739 0.355 0.043, 1.435 0.038

sigma2: 0.014

Ljung-Box Test: Q*= 2.991, degrees of freedom = 6, p-value = 0.810

Table 3: ARIMA model coefficients, p-values and Ljung-Box test results for overall opioids.

Model: Intercept + Ramp + Pulse + ARIMA (1, 0, 0)(1, 0, 2)

Coefficient Standard Error 95% Confidence Interval p-value

Intercept 6.110 0.221 5.678, 6.542 < 0.001

Ramp -0.055 0.049 -0.151, 0.040 0.256

Step 1.569 0.607 0.379, 2.759 0.010

Non-seasonal AR1 0.346 0.182 -0.010, 0.702 0.057

Seasonal AR1 -0.582 0.223 -1.019, -0.145 0.009

Seasonal MA1 0.463 0.229 0.015, 0.911 0.043

Seasonal MA2 0.643 0.258 0.138, 1.148 0.013

sigma2: 0.449

Ljung-Box Test: Q*= 1.495, degrees of freedom = 4, p-value = 0.828

Figure S2: ACF and PACF plot for overall Gabapentinoid

Furthermore, the presentation of the results, especially about pre- and post-intervention magnitude changes (descriptive stats) in the table, can be improved significantly by having explicit column headings (pre- and post-intervention) to report the absolute values and % changes. Please revise this.

Response: Thank you for this suggestion. We have reported the absolute values and clearly labeled pre- and post-intervention headings in Table 1. We agree that this improves clarity for readers.

Change:

Table 1: Percent changes and population adjusted sales volume per 1000 of gabapentinoid and opioid sales at 3-months, 6-months and 12-months following policy change.

Drug Class Pre-Intervention Population Adjusted Sales Post-Intervention Percent Changes (Population Adjusted Sales)

3-months 6-months 12-months

Overall Gabapentinoid 1831 -67.00% (604) -31.71% (1250) -24.35% (1385)

Gabapentin 197 +198.49% (588) +516.57% (1214) +583.00% (1345)

Pregabalin 1634 -99.00% (16) -97.81% (36) -97.57% (40)

Overall Opioid 6.12 +8.10% (6.61) +37.23% (8.40) +49.47% (9.15)

Similarly, a better way to report the ITS (segmented regression) is to fully report the results in a tabular format where the reader can appreciate the baseline (pre-intervention and post-intervention level and trend changes) by looking at coefficients and associated 95% CIs.

Response: Thank you for these suggestions, we agree that showcasing model output in tabular format will help readers. We added two additional tables showcasing the coefficients for the gabapentinoid and opioid ARIMA models, confidence intervals and p-values for the ARIMA analysis.

Change:

Table 2: ARIMA model coefficients, p-values and Ljung-Box test results for overall gabapentinoids.

Model: Intercept + Ramp + Pulse + ARIMA (0, 1, 0)(0, 0, 2)

Coefficient Standard Error 95% Confidence Interval p-value

Intercept 0.095 0.029 0.039, 0.151 < 0.001

Ramp -0.115 0.043 -0.199, -0.030 0.008

Pulse -0.894 0.065 -1.022, -0.766 < 0.001

Seasonal MA1 -0.310 0.130 -0.565, -0.054 0.017

Seasonal MA2 0.739 0.355 0.043, 1.435 0.038

sigma2: 0.014

Ljung-Box Test: Q*= 2.991, degrees of freedom = 6, p-value = 0.810

Table 3: ARIMA model coefficients, p-values and Ljung-Box test results for overall opioids.

Model: Intercept + Ramp + Pulse + ARIMA (1, 0, 0)(1, 0, 2)

Coefficient Standard Error 95% Confidence Interval p-value

Intercept 6.110 0.221 5.678, 6.542 < 0.001

Ramp -0.055 0.049 -0.151, 0.040 0.256

Step 1.569 0.607 0.379, 2.759 0.010

Non-seasonal AR1 0.346 0.182 -0.010, 0.702 0.057

Seasonal AR1 -0.582 0.223 -1.019, -0.145 0.009

Seasonal MA1 0.463 0.229 0.015, 0.911 0.043

Seasonal MA2 0.643 0.258 0.138, 1.148 0.013

sigma2: 0.449

Ljung-Box Test: Q*= 1.495, degrees of freedom = 4, p-value = 0.828

Finally, the graph representing the trend changes should also report counterfactual scenarios by dashed lines and 95% CI for the post-intervention trend line.

Response: Thank you for this suggestion. We have now included the graphs with the counterfactuals as Figures 3 and 4.

Change:

Figure 3: Overall gabapentinoid sales log per 1000 population with counterfactual scenarios by dashed lines and 95% CI for the post-intervention.

Author analysis based on IQVIA MIDAS® quarterly volume sales data for Egypt for the period Q1 2012 to Q3 2023, reflecting estimates of real-world activity. Copyright IQVIA. All rights reserved.

Figure 4: Overall opioid sales per 1000 population with counterfactual scenarios by dashed lines and 95% CI for the post-intervention.

Author analysis based on IQVIA MIDAS® quarterly volume sales data for Egypt for the period Q1 2012 to Q3 2023, reflecting estimates of real-world activity. Copyright IQVIA. All rights reserved.

Reviewer 2 Comments

1. The term “reclassification” in the title is ambiguous. Since pregabalin was designated as a controlled substance, it would be clearer if the title explicitly reflected this.

Response: Thank you for this suggestion. We have made appropriate changes to the title to improve clarity and specify that the reclassification we mention is about pregabalin being classified as a controlled substance.

Change: Impact of pregabalin reclassification as a controlled substance in Egypt on gabapentinoid and opioid utilization: a repeated cross-sectional study

2. The title should clearly specify what the “impact” refers to. Please revise the title to indicate that the study evaluates the impact on gabapentinoid and opioid utilization in Egypt.

Response: Thank you for this suggestion. We have revised the title to specify the impact.

Change: Impact of pregabalin reclassification as a controlled substance in Egypt on gabapentinoid and opioid utilization: a repeated cross-sectional study

3. The title specifies “repeated cross-sectional analysis,” but the design is more accurately described as an interrupted time series analysis using ARIMA models.

Response: Thank you for this suggestion. The STROBE Checklist for reporting of observational studies states that in the title we should “Indicate the study’s design with a commonly used term in the title or the abstract”. This suggestion indicates using the design, not the analysis, in the study title. Given this, we reported the study design of a repeated cross-sectional study rather than the analysis of an interrupted time series. We acknowledge that we used the term analysis in the title rather than study, which can be misleading, so we have made that change. If you would still like us to include the analysis used, we can add that into our title as well.

Change: Impact of pregabalin reclassification as a controlled substance in Egypt on gabapentinoid and opioid utilization: a repeated cross-sectional study

4. In the abstract, the terms “pulse,” “ramp,” and “step” are presented from a modeling perspective. It would be clearer to replace these with interpretation-oriented terms, such as immediate short-term effect, sustained level change, or gradual long-term trend change.

Response: Thank you for this comment. We have modified these terms in the abstract.

Change: ARIMA analyses of gabapentinoid sales showed a significant short-term effect (pulse: p < 0.001) and a notable gradual long-term change (ramp: p = 0.008). In contrast, opioids exhibited a significant short-term sustained increase (step: p = 0.010) but a non-significant gradual long-term change (ramp: p = 0.256), with sales rising by up to 49.5% at one year post-policy.

5. The description of IQVIA’s MIDAS database is insufficient for international readers who may not be familiar with it. More detail is needed on what the database is, how it is constructed, and what types of information it provides. In particular, please explain how drug information is captured and reported in MIDAS, and clarify whether this database can be considered representative for assessing medication use in Egypt. This additional context would help readers evaluate the validity of using MIDAS data for the study objectives.

Response: Thank you for this comment. We have added additional information about the IQVIA MIDAS data source and how information is captured. We also added additional information about the accuracy of the data and that it captures 92.6% of sales and is a good proxy for studying drug use in Egypt.

Change: IQVIA MIDAS reports monthly sales volume data from health systems, retail pharmacies, and major retail outlets (mass merchandisers, grocery stores, and convenience stores without pharmacies) for prescription and over-the-counter (OTC) drugs by country… An internal validation study by IQVIA reports a precision rate of 92.6% for Egypt of wholesaler, distributer and pharmacy audits. This indicates that it captures 92.6% of purchases made by drugstores wholesalers and manufacturers, suggesting sales volume is a good proxy for studying drug utilization.

6. The criteria by which the study data were selected from the MIDAS database should be described in the Methods section. Additionally, the results of this selection should be illustrated in a flow diagram and included in the Results section.

Response: Thank you for this comment. The data used is purchase data and is already aggregated at the national-level and, thus, identification at the pharmacy- or patient-level was not required as part of the steps. When pulling data from the MIDAS database, we filtered solely on drug name and formulation, and we have specified the drugs we have included in our methods. We did not have multiple steps when filtering for drugs, given this, we believe that a flow diagram will not help illustrate the drug selection process. However, we have made changes to the methods where we specify what “others” included. We believe this will provide transparency about the drugs included in the analysis.

Change: We used IQVIA MIDAS® quarterly volume sales data for the period Q1 2012 to Q3 2023 to analyse sales of oral and parenteral formulations of pregabalin, gabapentin and opioids (including codeine, fentanyl, hydromorphone, morphine, oxycodone, tramadol, nalbuphine and pethidine) in Egypt.

7. In the Methods section, the opioids are described as “including codeine, fentanyl, hydromorphone, morphine, oxycodone, tramadol, and others.” The use of “including” and “others” makes the definition unclear. A complete and explicit list of all opioid medications analyzed should be provided to ensure clarity and reproducibility.

Response: Thank you for pointing this out. We agree that specifying the drugs included in our study will improve clarity and reproducibility. We have specified what “others” included in the methods section.

Change: We used IQVIA MIDAS® quarterly volume sales data for the period Q1 2012 to Q3 2023 to analyse sales of oral and parenteral formulations of pregabalin, gabapentin and opioids (including codeine, fentanyl, hydromorphone, morphine, oxycodone, tramadol, nalbuphine and pethidine) in Egypt.

8. The description of the study model in the Methods section is not sufficiently detailed for reproducibility. More specific information on the intervention coding (step, pulse, ramp) and the final ARIMA specifications, including seasonal terms, should be provided. The ITS model should also be described in equation form, if available.

Response: Thank you for this suggestion, we agree that including this information will improve reproducibility. We have added the intervention coding to the supplement in Table S1, results from the Kwiatkowski root test in Table S2, and mentioned the ARIMA specifications in Tables 2 and 3 in the manuscript.

Change:

Table S1: ARIMA model intervention coding.

Step {1, year >= 2019.75; 0 elsewhere}

Pulse {1, year == 2019.75; 0 elsewhere}

Ramp {i, year >= 2019.75;0 elsewhere} where i is the number of periods(quarters) after third quarter of 2019.

Table S3. Kwiatkowski unit root test results.

kpss stat kpss pvalue

Gabapentinoid d = 0 1.23 0.01

Gabapentinoid d = 1 0.0868 0.1

Opioid d = 0 0.385 0.0838

Table 2: ARIMA model coefficients, p-values and Ljung-Box test results for overall gabapentinoids.

Model: Intercept + Ramp + Pulse + ARIMA (0, 1, 0)(0, 0, 2)

Coefficient Standard Error 95% Confidence Interval p-value

Intercept 0.095 0.029 0.039, 0.151 < 0.001

Ramp -0.115 0.043 -0.199, -0.030 0.008

Pulse -0.894 0.065 -1.022, -0.766 < 0.001

Seasonal MA1 -0.310 0.130 -0.565, -0.054 0.017

Seasonal MA2 0.739 0.355 0.043, 1.435 0.038

sigma2: 0.014

Ljung-Box Test: Q*= 2.991, degrees of freedom = 6, p-value = 0.810

Table 3: ARIMA model coefficients, p-values and Ljung-Box test results for overall opioids.

Model: Intercept + Ramp + Pulse + ARIMA (1, 0, 0)(1, 0, 2)

Coefficient Standard Error 95% Confidence Interval p-value

Intercept 6.110 0.221 5.678, 6.542 < 0.001

Ramp -0.055 0.049 -0.151, 0.040 0.256

Step 1.569 0.607 0.379, 2.759 0.010

Non-seasonal AR1 0.346 0.182 -0.010, 0.702 0.057

Seasonal AR1 -0.582 0.223 -1.019, -

---

## [Editor Report · Decision Letter 1]

14 Nov 2025

Impact of pregabalin reclassification as a controlled substance in Egypt on gabapentinoid and opioid utilization: a repeated cross-sectional study

PONE-D-25-13595R1

Dear Dr. Santhireswaran,

We’re pleased to inform you that your manuscript has been judged scientifically suitable for publication and will be formally accepted for publication once it meets all outstanding technical requirements.

Kind regards,

Claudia Garcia Serpa Osorio-de-Castro, Ph.D

Academic Editor

PLOS ONE
---

## [Editor Report · Acceptance letter]

PONE-D-25-13595R1

PLOS ONE

Dear Dr. Santhireswaran,

I'm pleased to inform you that your manuscript has been deemed suitable for publication in PLOS ONE. Congratulations! Your manuscript is now being handed over to our production team.

Kind regards,

on behalf of

Dr. Claudia Garcia Serpa Osorio-de-Castro

Academic Editor

PLOS ONE